# Robustness Disparities in Face Detection

**Samuel Dooley**[1], **George Z. Wei**[2], **Tom Goldstein**[1], **John P. Dickerson**[1]

[1]University of Maryland
[2]University of Massachusetts Amherst
{sdooley1, tomg, john}@cs.umd.edu, gzwei@umass.edu

## Abstract

Facial analysis systems have been deployed by large companies and critiqued by scholars and activists for the past decade. Many existing algorithmic audits examine the performance of these systems on later stage elements of facial analysis systems like facial recognition and age, emotion, or perceived gender prediction; however, a core component to these systems has been vastly understudied from a fairness perspective: face detection, sometimes called face localization. Since face detection is a pre-requisite step in facial analysis systems, the bias we observe in face detection will flow downstream to the other components like facial recognition and emotion prediction. Additionally, no prior work has focused on the robustness of these systems under various perturbations and corruptions, which leaves open the question of how various people are impacted by these phenomena. We present the first of its kind detailed benchmark of face detection systems, specifically examining the robustness to noise of commercial and academic models. We use both standard and recently released academic facial datasets to quantitatively analyze trends in face detection robustness. Across all the datasets and systems, we generally find that photos of individuals who are *masculine presenting*, *older*, of *darker skin type*, or have *dim lighting* are more susceptible to errors than their counterparts in other identities.

## 1 Introduction

Face detection identifies the presence and location of faces in images and video. In this work, face detection, also called face localization, refers to the task of placing a rectangle around the location of all faces in an image. Automated face detection is a core component of myriad systems—including *face recognition technologies* (FRT), wherein a detected face is matched against a database of faces, typically for identification or verification purposes. FRT-based systems are widely deployed [11, 33, 74]. Automated face recognition enables capabilities ranging from the relatively morally neutral (e.g., searching for photos on a personal phone [27]) to morally laden (e.g., widespread citizen surveillance [33], or target identification in warzones [51]). Legal and social norms regarding the usage of FRT are evolving [e.g., 28]. For example, in June 2021, the first county-wide ban on its use for policing [see, e.g., 24] went into effect in the US [29]. Some use cases for FRT will be deemed socially repugnant and thus be either legally or *de facto* banned from use; yet, it is likely that pervasive use of facial analysis will remain—albeit with more guardrails than today [67].

One such guardrail that has spurred positive, though insufficient, improvements and widespread attention is the use of benchmarks. For example, in late 2019, the US National Institute of Standards and Technology (NIST) adapted its venerable Face Recognition Vendor Test (FRVT) to explicitly include concerns for demographic effects [28], ensuring such concerns propagate into industry systems. Yet, differential treatment by FRT of groups has been known for at least a decade [e.g., 21, 41], and more recent work spearheaded by [8] uncovers unequal performance at the phenotypic

36th Conference on Neural Information Processing Systems (NeurIPS 2022) Track on Datasets and Benchmarks.

subgroup level. That latter work brought widespread public, and thus burgeoning regulatory, attention to bias in FRT [e.g., 39, 48].

One yet unexplored benchmark examines the bias present in a model's robustness (e.g., to noise, or to different lighting conditions), both in aggregate and with respect to different dimensions of the population on which it will be used. Many detection and recognition systems are not built in house, instead adapting an existing academic model or by making use of commercial cloud-based "ML as a Service" (MLaaS) platforms offered by tech giants such as Amazon, Microsoft, Google, Megvii, etc. With this in mind, our **main contribution** is a wide *robustness benchmark* of six different face detection models, three commercial-grade face detection systems (accessed via Amazon's Rekognition, Microsoft's Azure, and Google Cloud Platform's face detection APIs) and three high-performing academic face detection models (MogFace, TinaFace, and YOLO5Face). For fifteen types of realistic noise, and five levels of severity per type of noise [35], we test all models against images in each of four well-known datasets. Across these more than 5,000,000 noisy images from four commonly used academic datasets: Adience [20], Casual Conversations Dataset [34], MIAP [65], and UTKFace [82]. Additionally, to allow further research, we make our raw data available for exploration here: `https://dooleys.github.io/robustness/`.[1]

By benchmarking both commercial and academic models, we can understand two important insights: (1) audit the use-case of a company which takes open-source models to build in-house facial recognition models, and (2) adjudicate corporation's claims of caring about demographic biases in their products by measuring the extent to which their models are less biased than academic models which have no fairness considerations. As such, we endeavor to answer three research questions:

(**RQ1**): How robust are commercial and academic face detection models to natural types of noise?
(**RQ2**): Do face detection models have demographic disparities in their performance on natural noise robustness tasks?
(**RQ3**): Are the robustness disparities exhibited by commercial models more or less than the robustness disparities exhibited by academic models?

To answer these questions, we are motivated to understand how natural perturbations change the **system output.** We statistically analyze the performance of three common commercial facial detection providers and three state-of-the-art academic face detection models, comparing their performance and demographic disparities by comparing the output of the system on an unperturbed image with the output on a perturbed version of that image. This is interesting because it isolates the impact of the noise on the system, independent of the performance of the system. Thus, it makes comparing across systems easier. Focusing on **output** instead of system **performance** better isolates the impact of the stimulus of interest – the noise.

We observe that (RQ1) the leading face detection models show varying degrees of robustness to natural noise, but generally perform poorly on this task. Further, we conclude that (RQ2) these models do have demographic disparities which are statistically significant, and show a bias against individuals who are older, present as masculine, are darker skinned, and are dimly lit. Additionally, we see that (RQ3) these biases align with the commercial models, but that commercial model generally do not have lower level of disparity than the academic models.

Overall, our results suggest that regardless of a commercial company's commitments to equal treatment of different demographic groups, there are still pernicious problems with their products which treat demographic groups differently. We see further evidence that face detection is less robust to noise on older and masculine presenting individuals, which calls for future efforts to address this systemic problem. While our work indicates that the commercial providers are no worse on this important and socially impactful task than academics, we would hope to see that the commitments made by commercial companies would have them dedicate their vast resources and access to do better than comparatively under-resourced academics and substantially improve upon the robustness of their widely-used systems.

## 2    Related Work

We briefly overview additional related work in the two core areas addressed by our benchmark: robustness to noise and demographic disparity in facial detection and recognition. That latter point

---

[1]This work combines two unpublished papers which we wrote previously: [14] and [15]. This submission expands on those papers' ideas and enhances them with more rigorous analysis.

overlaps heavily with the fairness in machine learning literature; for additional coverage of that broader ecosystem and discussion around bias in machine learning writ large, we direct the reader to survey works due to [10] and [3].

**Demographic effects in facial detection and recognition.** The existence of differential performance of facial detection and recognition on groups and subgroups of populations has been explored in a variety of settings [8, 28, 37, 41, 56, 61]. In this work, we focus on *measuring* the impact of noise on a classification task, like that of [75]; indeed, a core focus of our benchmark is to *quantify* relative drops in performance conditioned on an input datapoint's membership in a particular group. We view our work as a *benchmark*, that is, it focuses on quantifying and measuring, decidedly not providing a new method to "fix" or otherwise mitigate issues of demographic inequity in a system. Toward that latter point, existing work on "fixing" unfair systems can be split into three (or, arguably, four [64]) focus areas: pre-, in-, and post-processing. Pre-processing work largely focuses on dataset curation and preprocessing [e.g., 22, 60, 63, 71]. In-processing often constrains the ML training method or optimization algorithm itself [e.g., 2, 12, 13, 26, 42, 52, 57, 71, 78, 79, 80], or focuses explicitly on so-called fair representation learning [e.g., 1, 5, 18, 19, 49, 72, 81]. Post-processing techniques adjust decisioning at inference time to align with quantitative fairness definitions [e.g., 32, 73].

**Robustness to noise.** Quantifying, and improving, the robustness to noise of face detection and recognition systems is a decades-old research challenge. Indeed, mature challenges like NIST's Facial Recognition Vendor Test (FRVT) have tested for robustness since the early 2000s [58]. We direct the reader to a comprehensive introduction to an earlier robustness challenge due to NIST [59]; that work describes many of the specific challenges faced by face detection and recognition systems, often grouped into Pose, Illumination, and Expression (PIE). It is known that commercial systems still suffer from degradation due to noise [e.g., 36]; none of this work also addresses the intersection of noise with bias, as we do.

Recently, *adversarial* attacks have been proposed that successfully break commercial face recognition systems [9, 66]; we note that our focus is on *natural* noise, as motivated by [35] with their ImageNet-C benchmark. Literature at the intersection of adversarial robustness and fairness is nascent and does not address commercial platforms [e.g., 54, 68]. To our knowledge, our work is the first systematic benchmark for commercial face detection systems that addresses, comprehensively, noise and its differential impact on (sub)groups of the population.

**Academic Face Detection Models.** Since 2012, neural-network-based face detectors have become ubiquitous in both industry and academia due to their comparative advantage in model capacity over traditional methods. As such, we are only going to focus on the prevailing approaches in deep face detection. According to Minaee et al. [53], there are five main categories of face detectors. *Cascade-CNN Based Models* generally use convolutional neural networks (CNNs) that operate at various resolutions to produce detections that are then repeatedly refined (or "cascaded") through non-maximum suppression and bounding box regression to ultimately output final face detections [44]. *R-CNN Based Models* utilize a region proposal network to predict face regions and landmarks and then verify that the candidate regions are faces or not with a Regional CNN [25]. *Single Shot Detector (SSD) Models* discretize the output space of bounding boxes over different aspect ratios as well as scales then use the confidence scores to reshape the default boxes to better contain the detected faces by using convolutional features from different layers, usually the higher level layers [47]. *Feature Pyramid Network (FPN) Based Models* upsample the convolutional features of higher (semantically richer) layers, aggregates them with those calculated in the initial forward pass to create semantically rich features at all image scales, then detects faces with each of these features at each layer [46]. *Transformers Based Models* use the Transformer [70] (or the Vision Transformer [16]) as the backbone for face detection. The academic models evaluated in this paper fall into the FPN or SSD based detector categories and were chosen because they were top performers of the popular WIDER FACE [76, 77] benchmark.

Our work is most closely related to that of [38], who look at *adversarial noise* and how that effects "gender detection", "age prediction", and "smile detection". [38] explicitly do not examine detection as defined by face localization, which is the topic of this study. Further, their facial analysis technologies generally are downstream processes from the facial detection/localization technology in this paper. Additionally, [50] provide a similar experimental design as our work though for face verification, and on a significantly smaller set of image distortions and test images. We refer the reader to [69] and [17] for surveys on bias in facial processing and biometrics.

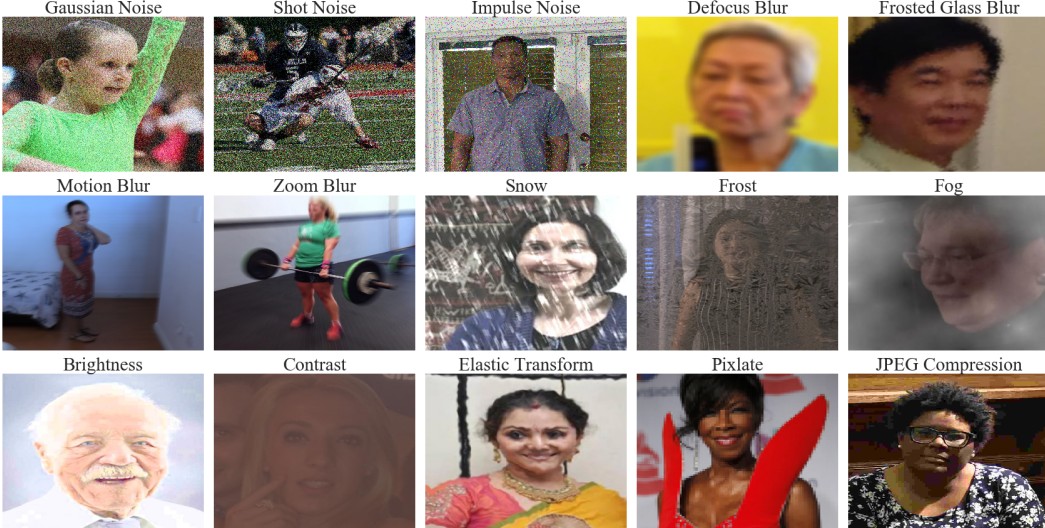

*Figure 1: Our benchmark consists of 5,066,312 images of the 15 types of algorithmically generated corruptions produced by ImageNet-C. We use data from four datasets (Adience, CCD, MIAP, and UTKFace) and present examples of corruptions from each dataset here.*

## 3 Benchmark Design

In this section, we outline the details of our benchmark by describing the data we used, the protocol or method we employed to answer out research questions, and the evaluation metric. We also describe how our benchmark can be used by other researchers, the limitations of our benchmark, and give an important social context for our study in facial analysis technology.

**Datasets** This benchmark uses four datasets to evaluate the robustness of three commercial and three academic face detection models. The datasets are described below.

The Open Images Dataset V6 – Extended; More Inclusive Annotations for People (**MIAP**) dataset [65] was released by Google in May 2021 as a extension of the popular, permissive-licensed Open Images Dataset specifically designed to improve annotations of humans. For each image, every human is exhaustively annotated with bounding boxes for the entirety of their person visible in the image. Each annotation also has perceived gender (Feminine/Masculine/Unknown) presentation and perceived age (Young, Middle, Old, Unknown) presentation.

The Casual Conversations Dataset (**CCD**) [34] was released by Facebook in April 2021 under limited license and includes videos of actors. Each actor consented to participate in an ML dataset and provided their self-identification of age and gender identity (coded as Female, Male, and Other), each actor's skin type was rated on the Fitzpatrick scale [23], and each video was rated for its ambient light quality. For our benchmark, we extracted one frame from each video.

The **Adience** dataset [20] under a CC license, includes cropped images of faces from images "in the wild". Each cropped image contains only one primary, centered face, and each face is annotated by an external evaluator for age and perceived gender (Female/Male). The ages are reported as member of 8 age range buckets: 0-2; 3-7; 8-14; 15-24; 25-35; 36-45; 46-59; 60+.

Finally, the **UTKFace** dataset [82] under a non-commercial license, contains images with one primary subject with annotated for age (continuous), perceived gender (Female/Male), and ethnicity (White/Black/Asian/Indian/Others) by an algorithm, then checked by human annotators.

For each of the datasets, we randomly selected a subset of images for our evaluation, with caps on the number of images from each intersectional identity equal to $1500$. This reduces the effect of highly imbalanced datasets. We include a total of 66,662 clean images with 14,919 images from Adience; 21,444 images from CCD; 8194 images from MIAP; and 22,105 images form UTKFace. The full breakdown of totals of images from each group can be found in Section A.2.

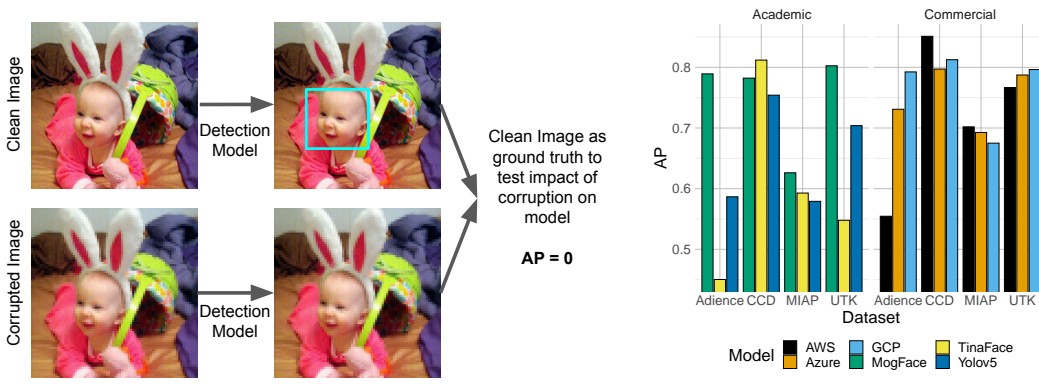

Figure 2: Depiction of how Average Precision (AP) metric is calculated by using clean image as ground truth.

Figure 3: Overall performance (AP) of each model on each dataset.

**Benchmark Protocol and Metrics.**    Recall, our motivating question is how the noise impacts a model's *output*. **To do this,** each image was corrupted a total of 75 times, per the ImageNet-C protocol with the main 15 corruptions each with 5 severity levels. Examples of these corruptions can be seen in Figure 1. This resulted in a total of $5,066,312$ images (including the original clean ones) which were each passed through each of the six models. Images were processed and stored within AWS's cloud using S3 and EC2. The experiments cost was $17,507.55 and a breakdown can be found in Appendix C.

We evaluate the change of the face systems under perturbations using the standard object detection metric: mean average precision (mAP). We use the standard implementation of the mAP metric by COCO [45]. Values reported below are mAP scores averaged over intersection over union (IoU) thresholds between 0.5 and 0.95 in intervals of 0.05. Below we call this metric Average Precision because we only have one class so the "mean" in mean average precision is trivial. Since we are interested in the system change under perturbation, and because none of the datasets have underlying ground truths, we treat the system output of the clean image as ground truth. A visual depiction of this process can be found in Figure 2.

We also investigate the significance of whether two groups are equally treated by a model under each metric by performing statistical tests. We observe bias by first performing a Kruskal-Wallis Rank Sum Test between explanatory and response variables which indicate whether two or more groups are treated equally or not. In the case where there is enough evidence to show that groups are treated differently, we then run the Pairwise Wilcoxon Rank Sum Tests to observe which groups have significantly different treatment and in which direction. All statistical tests are reported with $\alpha = 0.05$ with Bonferroni-Holm corrections. Each claim we make across datasets is done by looking at the trends in each dataset and are inherently qualitative.

We visually represent our results in Figures 4-7 by examining odds ratios between two categories of a sociodemographic variable across each model and dataset. For each pair of subgroups, like Middle-aged and Older subjects in Figure 5, we calculate the odds of each for each subgroup, $Odds_{middle}$ and $Odds_{older}$ and then look at their ratio: $Odds_{middle}/Odds_{older}$. When this value is greater than 1, like in Figure 5, it means the odds of higher performance are larger for middle aged group is higher than the older group. We conclude that there is a bias against older subjects. When the error bounds do not cross 1, this means that this disparity is statistically significant as well.

**How to Use our Benchmark.**    There are three main ways that our benchmark could be used by future researchers and practitioners. First, the analysis code, data, and results are being released publicly. New models that are built, either in academia or industry, can be easily benchmarked against our framework, and progress in this space can be tracked by the research community. Indeed, it is our intention to communicate our results to standards bodies such as NIST for inclusion in, or influence on, their long-running FRVT gauntlet. Second, the comparison across types of models (in our case, academic and commercial) could be adopted by more algorithmic audits. For example, in many areas (language models for text generation, diffusion models for text to image tasks, myriad object detection tasks) academic, industry-funded but open-sourced, and industry-funded and closed-source models compete across various metrics, and comparing and contrasting appropriately-defined bias metrics

across those verticals is of practical importance. Third, well-founded and quantitative studies may be of use to policymakers. As discussed in Section 1, facial analysis is a topic of great regulatory and legislative interest at this moment, and informing all sides—policymakers, the public, and providers of facial analysis technology—will lead to more clear and educated discussion and norm setting.

**What is not included in this study.** There are three main things that this benchmark does not address. First, we do not examine cause and effect. We report inferential statistics without discussion of what generates them. Second, we only examine the types of algorithmicaly generated natural-like noise present in the 15 corruptions. We explicitly do not study or measure robustness to other types of changes to images, for instance adversarial noise, camera dimensions, etc. Finally, we do not investigate algorithmic training. We do not assume any knowledge of how the commercial system was developed or what training procedure or data were used.

**Social Context.** This benchmark relies on socially constructed concepts of gender presentation and skin-tone/race and the related concept of age. While this benchmark analyzes phenotypal versions of these from metadata on ML datasets, it would be wrong to interpret our findings absent a social lens of what these demographic groups mean inside a society. We guide the reader to Benthall and Haynes [4] and Hanna et al. [31] for a look at these concepts for race in machine learning, and Hamidi et al. [30] and Keyes [40] for similar looks at gender.

## 4 Results

### 4.1 RQ1: Overall Model Performance

To answer RQ1 and to provide a baseline for comparison later in the analysis, we examine the overall performance of each model on each dataset, presented in Figure 3. We see from the outset that we can answer RQ1 affirmatively: face detection models sometimes struggle significantly with robustness to noise. Commercial models as a whole outperform the academic models on every dataset – however there are individual models in each category which break this conclusion. For example, the academic model MogFace performs significantly better than all the commerical models on UTKFace, though as a whole the academic models are inferior to the commercial ones.

Within in each class of model, commercial and academic, there is not a clear top model. However, we note that on the academic model side, MogFace significantly outperforms the other two models on every dataset except CCD. It is unknown as to why MogFace has such high performance, but we hypothesize a reason for what might explain this. MogFace was published very recently (late 2021), and perhaps much more recently than the commercial models. Only Azure indicates when its model was released (February 2021). The analysis of the commercial providers was also done prior to the release of MogFace. While more contemporary models do not necessarily imply better performance, this could be playing a role.

#### 4.1.1 Performance of Individual Perturbations

Recall that there are four types of ImageNet-C corruptions: noise, blur, weather, and digital. On Adience, Brightness is the easiest corruption and noise is the hardest on five of the six models – GCP performs best on Pixelate and worst on Snow. On CCD, all models perform best on Glass Blur but worst on Zoom Blur.

Again, he zoom blur corruption proves particularly difficult on the MIAP datasets – it is the worst performer on all models for this dataset, whereas Brightness is the easest on four of the six models. On UTKFace, elastic-transform is a notable corruption which the models struggle with – all models except TinaFace and Yolo5Face perform worst on elastic tansform and UTKFace; all models except GCP perform best on Brightness. TinaFace and Yolo5Face struggle very significantly with the noise corruptions on UTKFace. Further details and analysis can be found in Table 1 and Appendix D.2.

### 4.2 RQ2: Demographic Disparities in Noise Robustness

We now turn our attention to answer RQ2: do face detection models have demographic disparities in their performance on noise robustness tasks? Each dataset we analyze has both perceived gender and perceived age labels and CCD has perceived skin type and lighting conditions.

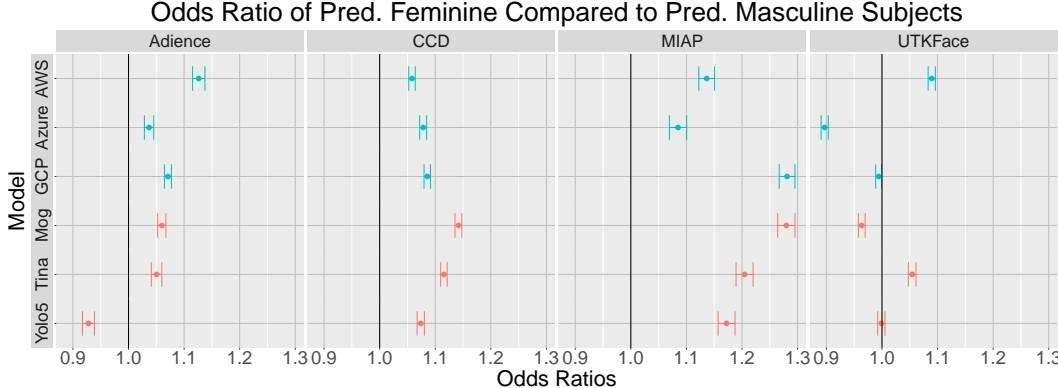

*Figure 4: Gender disparity plots for each dataset and model. Values below 1 indicate that predominantly feminine presenting subjects are more susceptible to noise-induced changes. Values above 1 indicate that predominantly masculine presenting subjects are are more susceptible to noise-induced changes. Error bars indicate 95% confidence.*

### 4.2.1 Gender Disparities

We begin by first pausing to note that the labels we have for perceived gender were in all cases provided by a third-party human reviewer, and the labels fall within the gender binary. The one exception is the MIAP dataset which reports a category of "Unknown" for times when the human reviewers were unable to reach a decision on the perceived gender of the subject. While gender is not binary and gender identity is not something which third party reviewers can infer, we use the perceived gender concept in our work to measure how model performance may differ for people who present gender differently.

We visually depict the performance of each model on each dataset in Figure 4 broken down by perceived gender. We analyze the observed perceived gender disparities for each dataset separately with a report of the odds ratio of feminine presenting individuals over masculine presenting individuals. Recall, values over 1 indicate higher performance on those whose are feminine presenting, and values less than 1 indicate higher performance on those who are masculine presenting.

We observe, qualitatively, across the 24 dataset and model combinations, there is a bias against masculine presenting individuals in 19 of them, there is a bias against feminine presenting subjects in 4, and there is no bias in one. This is a rather surprising result as previous reports indicate biases against feminine presenting individuals in facial recognition technology.

We further observe that the UTKFace dataset has the lowest robustness bias for perceived gender across all the models. This indicates that the dataset itself is an important tool in the measurement of algorithmic disparities and suggests that future work in this domain area should greatly expand their studies to incorporate multiple datasets.

### 4.2.2 Age Disparities

We move on to a discussion of the age disparities present in these models and datasets. We report the results of this age disparity in Figure 5. We note again, that age labels are given by perceived age of the subject in the image. Adience provides disparate age categories, MIAP provides age groupings (Young, Middle, Older, and Unknown) and UTKFace natively provides a numeric value. Since numeric age values from UTKFace are likely misspecified as it is nearly impossible to correctly predict a person's age from a photo, we bin these numeric values into four buckets of (0-18), (19-45), (45-65) and (65+).

Qualitatively, looking at all these results, we observe that the oldest group always is more susceptible to noise-induced changes compared to middle aged individuals. Quantiatively as well, we see that the oldest group is always statistically significantly the lowest performer of the groups. We note that while there may be differences in the sample sizes of these groups, the statistical tests are robust to these differences and account for sample size differences. Statistical test results for Pairwise Wilcoxon Rank Sum Tests can be found in the Appendix.

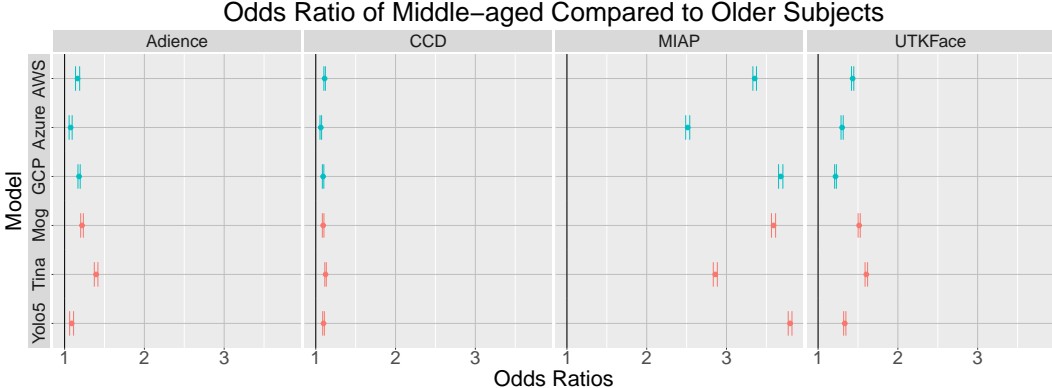

*Figure 5: Age disparity plots for each dataset and model. Values greater than 1 indicate that older subjects are more susceptible to noise-induced changes compared to middle aged subjects. Error bars indicate 95% confidence.*

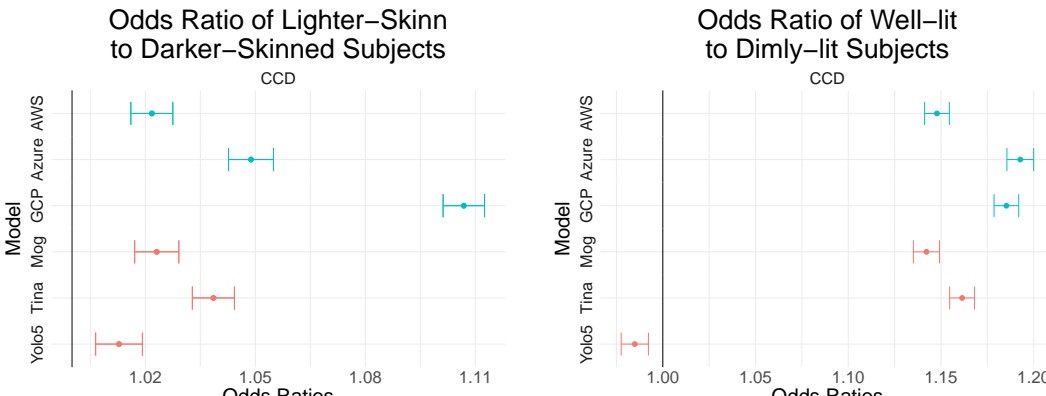

*Figure 6: Skin type disparity plots for CCD. Values above 1 indicate that darker-skinned subjects are more susceptible to noise-induced changes. Error bars indicate 95% confidence.*

*Figure 7: Lighting disparity plots for CCD. Values above 1 indicate that dimly-lit subjects are more susceptible to noise-induced changes. Error bars indicate 95% confidence.*

For MIAP, we observe significantly higher biases against older individuals than we do for the other datasets. We hypothesize that this might be due to the way in which the MIAP dataset was collected and the nature of the more natural images of entire scenes with sometimes multiple faces in them.

### 4.2.3 Skin Type and Lighting Disparities

The only dataset which includes metadata on skin type and illumination is the CCD dataset. As was customary at the time of the dataset release, CCD reports annotator provided Fitzpatrick skin type labels which we split the into two groups: Lighter (for ratings I-III) and Darker for ratings (IV-VI).

We observe a statistically significant bias against dark skinned individuals across every model, as can be seen in Figure 6. We further report that the bias between skin types is highest in the youngest groups; and this bias decreases in older groups. We also see a similar trend in the intersectional identities available in the CCD metadata (age, perceived gender, and skin type). We see that in every identity (except for 45-64 year old and Other gendered) the darker skin type has statistically significant lower AP. This difference is particularly stark in 19-45 year old, masculine subjects.

Lighting condition is also included as a metadata label in the CCD dataset. In Figure 7, we see that every model, except for YOLO5Face, exhibits behavior such that dimly lit images are more susceptible to noise-induced changes than brightly lit images. Interestingly and across the board, we generally see that the disparity in demographic groups decreases between bright and dimly lit environments. For example, the difference in precision between dark and light skinned subjects decreases to zero in dimly lit environments. This is also true for age groups. However, this is not true for individuals with gender presentations as Other or omitted.

### 4.3 RQ3: Disparity Comparison to Between Academic and Commercial Models

To answer RQ3, we examine the ordering and overlapping of the confidence intervals in the Figures 4-7. We note that we do not see signs of systemic differences between academic and commercial models in terms of their demographic disparities. When we examine the most biased model in each of the dataset and sociodemographic pairings, we observe no clear pattern. Commercial models are most biased in skin type and lighting variables as well as on Adience and UTKFace in the perceived gender variable. Academic models are most biased on the CCD dataset in the perceived gender variable as well as every dataset except for CCD on the age variable. (They are tied in the other two instances). Thus, we conclude there is no systematic difference in the magnitude of the disparity exhibited by commercial and academic models writ large.

## 5    Implications and Hypotheses

Above, we have shown striking disparities in commercial facial analysis systems. These biases have potential for real harms felt by individuals. Facial *detection* is the first step in facial recognition. As such, the biases which we report here will propagate downstream into further facial analysis systems. Facial detection bias is the starting point for bias in other facial analyses, and research that addresses biases in detection will also serve any other facial analysis system which uses its outputs. However, downstream systems will still have their own biases.

Since we are external researchers, we can only speculate as to why these disparities and biases are observed since we do not have access to the models themselves. The biases for dark-skinned individuals and dimly-lit subjects is unfortunately aligned with many prior works on the subject. Among the reasons for this include luminance and pixel intensity, which unfortunately have been codified as being discriminatory against darker skinned people in photography for decades [43].

On the other hand, the findings about older individuals and masculine-presenting individuals offer contrasting conclusions from existing work that audits facial analysis technologies. Regarding the finding the systems are more susceptible to noise-induced changes on masculine presenting subjects, we hypothesize that this might have to do with the size that a feminine-presenting subject's *head* takes up in an image. One gender presentation marker is hair and we hypothesize that the subject's entire head size might be a confounding factor in this bias phenomenon. We unfortunately do not have the data to test this hypothesis (since ground truth data for face detection includes just data on the face), but one could collect such data with sufficient ground truth.

## 6    Discussion & A Call to action

Revisiting our research questions, we come away with rather clear answers. We see that face detection models:

(**RQ1**):  show that their robustness to noise could be improved significantly;
(**RQ2**):  have significant perceived sociodemographic disparities in their performance on noise robustness tasks; and
(**RQ3**):  show similar degrees of demographic bias across both academic and commercial models.

We believe that these results beget three main conclusions for different audiences who are interested in face detection systems and/or algorithmic bias. Our results suggest that commercial systems generally are no less biased on noise robustness than academic systems, for the types of noise corruptions we benchmarked. This is a rather striking result considering the resources large companies have at their disposal to tackle problems like demographic disparities in their products. Additionally, since demographic disparities in commercial products became a crucible following the publication of Buolamwini and Gebru [7] in 2018, these corporations have had ample time to address and work towards solutions to these issues. While these companies have to varying degrees acknowledged the need to equal out demographic disparities in their products, we cannot fully know what investment they have placed on these issues, and specifically on disparities in noise robustness. So at this time, we can merely speculate.

If these companies have committed vast resources to address the demographic disparities in their products, and specifically in noise robustness, then our results lead us to conclude that these investments have generally not paid off. We conclude this because we now know that within each dataset and for most commercial model, there is at least one academic model which is at most as

biased than it is. Further, since these academic models are published publicly with full source code and training procedures, we know that these models have not included any fairness constraints or considerations. Thus, if these companies *have* invested heavily in this problem, then we conclude that their investments have not paid off.

However, it is perhaps overly optimistic to think that corporations have invested in the mitigation of demographic bias in noise robustness — although we posit that this is not likely because many real-world use cases for facial analysis occur under imperfect "in-the-wild" conditions that would introduce various forms of natural noise. If in fact they have not done so, our results give a clear benchmark and goalpost for these corporations to improve. While in most cases, the commercial models are the most biased system, we should endeavor to expect that if these corporations plan to continue to publicly sell face detection software — a very socially and ethically provocative tool — that they should be investing in mitigating these biases and be able to do better than academic models which have no fairness considerations.

Our results add to the increasing body of research which finds various pernicious forms of demographic bias in facial recognition technologies. We provided strong evidence of the demographic biases present in face detection systems. We conclude that despite all the talk and publicity about concerns of demographic disparities in commercially provide products, large technology companies are no better at eliminating bias for noise robustness than academic models. Thus, we end this work with two broad calls to action:

**To industry:**  This benchmark shows that the highly-resourced companies are no better than academic models at this robustness disparity in facial detection, a rather surprising comparison between where a trillion-dollar company could be—by spending a vanishing fraction of their liquid capital—and where it *should* be—where "should" is, admittedly, a value judgment, but a bipartisan one [6], and one gaining increasing traction in those firms' own home country [62].

Our call to action, then, is as follows: pay attention to, work with, and fund academic research in unfairness in facial detection and noise, specifically natural and synthetic styles of noise. As our present work shows, academic models run hand-in-hand with—and, indeed, by some metrics beat—commercially deployed systems, and it would be of great benefit to further encourage unrestricted growth in that space, and to fertilize that growth with cross-boundary communication of techniques that have been tried internally at for-profit firms. Specific to our setting, both the present work and previous works [e.g., 7, 61] would benefit immensely from at least partial access to the internal workings of commercial systems, including dataset curation processes. Beyond simply measuring disparities, the natural next step is to hypothesize reasons for those disparities and then to, at least partially, mitigate them via new techniques. Indeed, as this paper shows, state-of-the-art academic models are arguably *beating* commercial models in some ways, so the value within this communication would flow both ways. Without a clear line of communication between academic and industrial researchers, this latter process is hampered.

**To the public sector:**  The public sector provides a great service in both impacting the evolution of, and creating as well as enforcing the present state of social and legal norms. For example, in the United States, for our specific setting, the National Institute of Standards and Technology (NIST) Face Recognition Vendor Test (FRVT) has measured and monitored progress in both commercial and academic facial analysis systems. It has been run for at least the last two decades, and has been updated numerous times. Indeed, in a recent FRVT Update, NISTIR 8280 (2019), NIST brought demographic concerns into the forefront. NIST's venerable FRVT has a history of incorporating natural noise into its barrage of tests; we would ask NIST, and analogous non-regulatory and standards-settings bodies in other countries, to consider updating their tests (e.g., FRVT) to include the cross section of bias and forms of noise. Our work motivates the need for monitoring in this area.

To the regulatory side, we are encouraged by and seek further acceptance of results publicized by both academics and industrial researchers. Washington State aims to set an example here with its recently enacted State Bill 6281, which states "if the results of . . . independent testing identify material unfair performance differences across subpopulations . . . then the processor must develop and implement a plan to address the identified performance differences" [55]. We believe that this benchmark meets this definition and hope the public sector has a robust enforcement mechanism for such legislation. We encourage other researchers to continue to audit existing commercial products, and believe our approach to compare commercial models to academic models enriches the scholarly and social discourse about facial recognition technology.

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
