# OpenReview forum: "Robustness Disparities in Face Detection"
_NeurIPS.cc/2022/Track/Datasets_and_Benchmarks — NeurIPS 2022 Datasets and Benchmarks _

### Official Review · Reviewer_eCuq · 2022-07-18
**A benchmark on demographic disparities of face detection on natural noise robustness tasks**

**Rating:** 4
**Confidence:** 4
**Correctness:** The evaluation methods and experiment…

**Strengths:**

1.	The authors conducted comprehensive experiments on different models, datasets, protected attributes, and types of noise.
2.	The authors provided detailed analysis of the experiment results.
3.	The authors provided some hypotheses to explain the observation.

**Weaknesses:**

1.	Unfair comparison: It’s known that training data is essential for better understanding the behavior of models. Although the authors cannot control or investigate the training process of the commercial models, the training data of academic models should be the same for a fair comparison.
2.	I would like to see hypotheses for explaining the observations but as the authors pointed out, there are few experiments to evaluate the proposed hypotheses. So it's still a weakness nonetheless.

**Additional Feedback:**

1. Missing refs in Line 199 & Line 288.

**Clarity:**

The paper is overall well-written.
But I think there might be some missing refs or mislabeled figures that kind of threw off my ability to fully understand a few things??

**Documentation:**

The Github repo is documented well enough to reproduce the datasets and re-run the experiments done for the paper submission.
But I did not see a license on GitHub.

**Ethics:**

No concerns to raise.

**Relation To Prior Work:**

This paper discussed enough related work and the difference from previous works.

**Summary And Contributions:**

This paper investigates the disparity in the robustness of face detection models to natural noise. Experiments on 6 different models (3 commercial models and 3 academic models) and 4 datasets under 15 different types of noise have been done and the authors empirically show that photos of individuals who are masculine presenting, older, of darker skin type, or have dim lighting are more susceptible to errors than their counterparts in other identities.  Some hypothesis for explaining the observation is proposed without empirical verification.

---

> ### Author Response · Authors · 2022-08-29
> **Following up with Reviewer eCuq**
>
> Thank you again for your thoughtful review. Does our response help address your feedback? We would appreciate the opportunity to engage further if needed.

---

### Official Review · Reviewer_XZC1 · 2022-07-22
**This paper proposes a benchmark to evaluate academic and commercial Face Detection Models using “natural” noise perturbations.**

**Rating:** 6
**Confidence:** 5

**Strengths:**

The paper’s contributions are extensive in size due to the scale of experiments- 75 different
perturbations (15 perturbations * 5 severity levels for each), 4 datasaets, 6 models. This will be
helpful for future researchers. The paper is relevant to the current needs of the research community
as it adds to a growing line of work in the domain of third-party audits for Facial Recognition
Technology. The authors provide access to the raw data and their codes to reproduce the
experimental results. The paper takes care to address the social and ethical considerations
surrounding gender and age in their experimental results.

**Weaknesses:**

The paper has multiple weaknesses as highlighted below-
1. This paper is not the first contribution towards studying robustness of Face Detection
Models under adversarial conditions. The authors should see Jaiswal, Siddharth, et al. &quot;Two-
Face: Adversarial Audit of Commercial Face Recognition Systems.&quot; Proceedings of the
International AAAI Conference on Web and Social Media. Vol. 16. 2022. [The arxiv version is
available since Nov 2021]] which evaluates commercial Face Recognition models under
adversarial natural noise perturbations. The reference to this very importart to prior art and subsequent comparison is completely missing.
2. The authors do not calculate the accuracy on the original set of images and thus there is no
way to know how good the models are performing on those images. The relative comparison
between the original and perturbed images might not be fair if the original accuracy itself is
low.
3. The authors do not present any comparison between the different forms of noises and thus
there is no way to know which perturbation is causing the most bias and which is the most
robust model for a given perturbation. This is a serious weakness.
4. On Page 4, Section 3, last paragraph- “We visual represent …” should be “We visually
represent”
5. On Page 5, Section 4.1, first paragraph- The reference to Figure 3 is missing.
6. On Page 5, Section 4.1- The authors state that MogFace may be using Adience and UTKFace
datasets for their training, but no such claim is made in the MogFace paper. The authors
should retract this claim, rather stating that the reason is unknown or verify the reason.
7. On Page 6,7, Figures 4-7- The individual figure boundaries are not clear. Either each figure
should be in a separate plot area or there should be clear vertical separation between the
figures.
8. On Page 6, Section 4.2.1, third paragraph- The authors state that previous works indicate
bias against feminine presenting individuals—This could be due to the datasets used in those
papers and hence a generalized statement may not be valid. This can be only confirmed if
the results are compared for the same dataset over time.
9. On Page 7, Section 4.2.2, second paragraph- “Qualitatively, looking at all these data…”
should be “Qualitatively, looking at all these results…”
10. On Page 8, Section 4.3, first paragraph- “Thus, we conclude there is not systematic
difference …” should be “Thus, we conclude there is no systematic difference…”
11. On Page 8, Section 5, first paragraph- Reference to a section is missing.
12. On Page 8, Section 6, second paragraph- The authors state that commercial models are no
less biased on noise robustness but they have only tested for naturally occurring noises. The
distinction should be clarified in the text.
13. On Page 9, Section 6, first paragraph- The authors claim that for every commercial model,
there is an academic model that is less biased than it—this claim is untrue for a given
dataset as inter-dataset comparison is unfair due to the difference in images. In the next
paragraph, the authors claim that the commercial models are not the biased model. These
are conflicting and the authors should resolve this.
14. On Page 9, last paragraph- “approach to compare commercial biases to academic biases”
should be “approach to compare commercial models to academic models” as the biases
being studied are the same.

**Additional Feedback:**

None.

**Clarity:**

The paper is written clearly with a few grammatical errors, which have been already pointed out.

**Correctness:**

While the claims in the submission concerning the results are correct, but the claim of being a
unique work in this domain is not true as already informed earlier. The datasets for perturbations
are constructed using standard ways. The evaluation methods and experiment design is not very
sound as the authors don’t compare between the different perturbations and thus there is no way
to determine which model is robust to which form of noise and whether the bias for certain types of
noise is higher or not. Similarly, the authors do not present any results on the accuracy for the
original images, instead assuming that to be the ground truth. This may not work if the actual
accuracy itself is low.

**Documentation:**

Yes, the documentation is sound and there is sufficient detail to support reproducibility.

**Ethics:**

There are no ethical concerns as the datasets, models and perturbations are all cited from existing
literature.

**Relation To Prior Work:**

The authors make a sound attempt to distinguish their contribution from the existing work but miss
out on one important citation which significantly weakens their work. The authors should perform a
stronger literature survey and compare how their work differs from Jaiswal et. al’s work.

**Summary And Contributions:**

Summary: The authors have proposed a benchmark for 6 Face Recognition Systems- 3 commercial
and 3 academic which are tested using 15 different natural noise perturbations from 4 datasets. The
scope of the experiments is extensive, and the results are interesting—the authors show that there
is a clear bias against masculine presenting, older individuals who either have dark skin or are
present in dim lighting. The authors conduct statistical significance tests to verify the correctness of
their results and the experiments are sound.
Contributions: The authors have contributed a new work to the growing line of audit studies for Face
Recognition Systems by testing existing commercial and academic systems for biases using multiple
datasets and perturbation techniques. They also make their raw data and codes available for public
use.

---

### Official Review · Reviewer_tpDR · 2022-07-26
**Effect of synthetic noise on face detection using existing datasets and models in the context of fairness**

**Rating:** 7
**Confidence:** 5

**Strengths:**

1. The authors select three academic and three commercial face detection models, covering a good range of models.
2. The authors select four databases containing a diverse set of images.
3. The work is well motivated and addresses an important issue.
4. Overall, the paper is well written.

**Weaknesses:**

The major concern is the relevance of the work in the Dataset and Benchmarks Track. There is no dataset proposed for the study. The paper focuses on experimental evaluation. More emphasis on how this benchmark is designed and can be used in the future needs to be provided.

**Additional Feedback:**

1. The study does not propose a novel dataset. Therefore, an effort is required to modify the paper writing to fit better into the requirements for this track. (Elaborated in the 'Weakness' section)
2. There are references to Figures missing in the main paper as well as the supplemental material. Ex- Line 288 of the main paper.
3. Some related work is missing. Relevant papers have been suggested in 'Relation to Prior Work.'
4. The authors specify that the MogFace dataset has been trained on the Adience and UTKFace datasets. This makes it unfit to be used for evaluation in the benchmark unless the authors ensure there is no overlap in data during evaluation.

**Clarity:**

Yes, the paper is well written. However, more emphasis is required on the design and use of benchmarks for this track.

**Correctness:**

The experimental design and evaluation methods appear to be thorough and correct.

**Documentation:**

The datasets and models have been described in the main text. Information from Appendix A can be incorporated into the main paper for further detail. No dataset is proposed in the study. Therefore, there are no concerns regarding availability, maintenance, ethical and responsible use. Reproducibility for the study is supported by the GitHub link shared by the authors.

**Relation To Prior Work:**

Yes, the difference from previous work is discussed.

The following relevant literature should be discussed.
1. Majumdar, Puspita, et al. "Unravelling the Effect of Image Distortions for Biased Prediction of Pre-trained Face Recognition Models." Proceedings of the IEEE/CVF International Conference on Computer Vision. 2021.

The following surveys on bias in facial analysis may be added.
1. Singh, Richa, et al. "Anatomizing bias in facial analysis." Proceedings of the AAAI Conference on Artificial Intelligence. Vol. 36. No. 11. 2022.
2. Drozdowski, Pawel, et al. "Demographic bias in biometrics: A survey on an emerging challenge." IEEE Transactions on Technology and Society 1.2 (2020): 89-103.


**Summary And Contributions:**

The authors study the effect of noise in existing face detection models- commercial and academic. They specifically study demographic bias for gender and age-based subgroups. The performance of academic and commercial models is evaluated using four existing datasets, and the corresponding observations have been provided.

---

### Official Review · Reviewer_QktW · 2022-07-26
**Strong Paper highlighting Biases of Facial Detection Algorithms**

**Rating:** 8
**Confidence:** 2
**Clarity:** The paper is well written and the fin…

**Strengths:**

- The paper combines techniques used in analysis of facial recognition models within a new context that was previously under-researched
- The authors show a clear understanding of the limitations of their approach and are careful about the framing and conclusions drawn for the most part. For instance using annotations from third parties at face value can often lead to dangerous assertions but the authors were careful to frame the annotations as presentational observations which is rooted in the societal context of the problem they are trying to address
- The authors' steps and their reasoning is clearly communicated to the readers
- The results are a significant contribution to the ongoing discussion within discussions of biases in facial analysis algorithms. Furthermore they particularly show that commercial algoritms are no less biased than academic algorithms, which is an important distinction.

**Weaknesses:**

- The authors justify their scope of analysis being neural networks due to higher accuracy of neural networks compared to previously used methods. However this does ignore that outside of Machine Learning as a Service platforms and labs at top ranked universities, traditional HAAR Cascade Classifiers are still extremely popular due to the lack of computational resources required. Considering that it would not have been too resource intensive to repeat the experiments with cascade classifiers, it is disappointing that the authors did not choose to perform them to add an additional baseline benchmark to compare their other results with.
- Aside from robustness comparisons, it would have been extremely useful to have comparisons of the algorithms using the clean images such as likelihood of a face being detected across gender presentations, age presentations and skin tones without noise corruption. The performance of the algorithms within the ideal conditions is also an important baseline to be established.

**Additional Feedback:**

- Please correct some of the broken references: opening paragraphs of section 4.1 and section 5
- The discussion might also include additional disscussions borrowing from social sciences regarding visibility, and desire for marginalized groups to be seen or observed or lack thereof.

**Correctness:**

The paper is constructed well and the dataset collection and evaluation steps are easy to follow and logical.

**Documentation:**

There is considerable detail on dataset collection, in addition to external links to the dataset.

**Ethics:**

The authors are careful to frame their findings as relating to gender presentation and age presentation and do not make sweeping statements about gender identity, age, ethnicity and race despite some of the datasets used doing just that. Furthermore the paper is largely focused with uncovering biases that would potentially large ethical ramifications and targetted harm to individuals.

**Relation To Prior Work:**

The paper is well situated as following up to previous work in analysis of biases in facial recognition and the authors clearly position their paper's contribution to the broader discussion. In particular the authors mention the time elapsed since seminal works within the field and how there has not been significant improvement in the performance of commercial facial detection applications since then.

**Summary And Contributions:**

The paper makes the claim that analysis regarding biases in facial recognition system has largely ignored an equally important prerequisite step, which  automated face detection. The paper attempts to contribute towards this research gap by conducting experiments for robustness under naturalistic noise corruption on six facial detection algorithms, three from commercial applications and three from academic The authors use data from four different datasets that are annotated by third parties with labels for gender presentation and age presentation, with only one of the datasets being annotated for skin tone and lighting.The paper concludes firstly that subjects with darker skin tone and in dim lighting are more likely to have errors when subject to noise corruption, which is in-line with previous research on algorithmic bias of facial recognition systems. The paper also finds the masculine presenting faces and older presenting faces are also more prone to error, which is a novel finding.

---

### Official Review · Reviewer_cekZ · 2022-07-28
**A benchmark for face detection must be clear on the task and the images**

**Rating:** 4
**Confidence:** 4
**Clarity:** The paper is reasonably clear, except…

**Strengths:**

- The dataset is suitably large, with tens of thousands of images (original images) enhanced into a set of 5 million images.
- The dataset contains many image transformations, some of which are important with respect to real-world performance of face-processing technologies.
- The paper uses the benchmark to assess six face-recognition systems, including several widely adopted systems.
- The supplement and supporting materials thoroughly document the benchmark and its application.

**Weaknesses:**

When developing a benchmark for a face-processing technology, three key desiderata are a clear description of (1) the task that the systems will perform, (2) the image set that the systems will be tested on, and (3) the performance criterion on which the systems will be judged. I see weaknesses in the authors' description of each:

1 & 3. The task and performance criterion.

Though often conflated in marketing materials about face-processing technologies, and sometimes even in ML papers on the topic, there are two distinct tasks called "detection". The first task is the traditional detection task developed by psychophysicists to study human visual processing: determining whether a face is present or absent in the image. The second task is the task widely studied in computer vision: face localization, which requires the system to locate one or more faces present in the image, perhaps by drawing a bounding box around each or place markers at predefined facial landmarks. And occasionally there is a third, hybrid task that is used, which requires the system to located zero or more faces present in the image, returning some null value when no faces are detected. It is this second (or possibly third) task that the authors study.

However, the authors use an average precision metric that applies to binary or multilabel classification tasks (see lines 162–167), where, rather than computing the typical metric of performance on face localization (IoU, intersection over union), the authors instead compute IoU, set a threshold IoU (over a range with unstated discretization/levels), use it to binarize the outcome, and compute the average precision with respect to that. The reason for this roundabout metric of performance is unclear, and made all the more confusing by the fact that some of the tested face-detection systems in fact provide confidence scores in the detection that could be used directly when computing metrics derived from ROC curves.

2. The image set.

Given that the proposed image set aggregates and then enhances four existing image sets, the contribution here must be judged on the extent to which the enhancements lead to an image set that is better suited to gauging disparities in the performance of face-detection systems. Otherwise the contribution (with respect to benchmark design) would amount to testing the performance of existing algorithms on existing datasets on existing tasks, leaving the novelty here to the assessment of these particular algorithms and what was found, which is interesting and well worth pursuing as a line of research, but is not a new benchmark or dataset and less of an advance than it might be were the image set itself a contribution.

The proposed enhancements transform the original images through 15 different image manipulations that the authors refer to as "noise".
(I do not see a definition of "noise" anywhere in the paper, and it is used in a non-standard way.) What's sorely needed here is (A) a justification for why these 15 image manipulations are appropriate for the authors' aims, and (B) a performance analysis with respect to the original image sets vs. each of these manipulations in turn. Several of the manipulations, such as Gaussian noise, defocus blur, JPEG compression, and motion blur, are perhaps close enough to real-world effects of low lighting, failing to get the subject in focus, and moving the camera during exposure, that it may be possible to argue that the authors are simulating real-world image quality issues. However, other manipulations are harder to justify. And even if they were justifiable, no attempt is made to determine the level of these image quality issues in the original source material, so that for example, two images might have equal levels of defocus, one because the original image had poor focus, the other because the original image had good focus but defocus blur was added. This concern about the image set goes to the core of the paper's contribution: what is the significance of determining the robustness of face detection systems to artificially introduced image transformations vs. the kinds of image quality issues that are already present in large-scale image datasets that are widely available?


**Additional Feedback:**

Line 176, typo, "We visual represent" => "We visually represent"

Line 348, I did not understand this call to action. Industry already pays attention to, works with, and funds academics research in this space. For example, Amazon employees attend NeurIPS, funds faculty research through several grant programs, hires people with dual appointments at universities, and even creates new products aimed at assessing biases of the kind presented in the manuscript via SageMaker Clarify.

**Correctness:**

The evaluation methods and experiment design are appropriate and performed correctly, though see concerns about evaluation methods above.

**Documentation:**

Yes, there is sufficient detail to support reproducibility. In general, the level of documentation is a strength of this manuscript. My only gripe is that, for the commercial systems, it is (understandably) difficult to provide enough detail for reproducibility because the underlying systems are not generally available and may change over time. As such, I recommend the authors note the exact date on which the systems were tested, which provides at least some light versioning.

**Ethics:**

The submission appears to follow the guidelines.

On line 223, the authors write "…gender identity is not something which third party reviewers can assess." This statement is insufficiently precise. Many individuals aim to signal their gender identity by modifying their faces through processes such as depilation, plastic surgery, makeup application, facial hair and beard styling, and facial posture. It would be odd to say that in such communicative contexts, where a third-party correctly interprets a gender identity that is deliberately signaled by the person holding that identity, that the third-party has not "assessed" (or, I'd prefer to say, "inferred") that individual's gender identity.

The disclaimer of sorts at line 225 should go at the first mention. I'd have preferred the authors to be crystal clear about the distinction on every mention and not resort to a shorthand at any point.

One lines 233, the "bias against" language is insufficiently precise. The ethical implications of disparate performance of a deployed face-recognition technology on a task depends on the context in which the technology is deployed and how the disparate performance is mitigated. For example, consider a disparity found in performance of a surveillance system with respect to a marginalized group. No justice would be found in correcting the disparity by improving the performance of the surveillance system to better localize or recognize members of that group. In a domain such as face-recognition, where there is widespread societal debate about their use and misuse, one must be careful when discussing what is a bias "for" or "against". The language in the abstract is a bit better in this regard.


**Relation To Prior Work:**

It is unclear how the current approach relates to existing benchmarks such as performance on Labeled Faces in the Wild or the many other datasets that reproduce images under conditions where the "noise" added by the authors was introduced at the time the images were taken, not post-hoc.


**Summary And Contributions:**

In this submission, the authors introduce a new benchmark for assessing disparities in face-detection systems caused by image perturbations introduced into images ("noise"). The benchmark involves measuring performance in a face-detection task across four new image sets derived from adding noise to previously released image sets. The authors then benchmark 6 face-detection algorithms, 3 released by industry and 3 released by academics. Next the authors interpret the performance of these algorithms with respect to disparities across groups defined by perceptions of gender, age, skin type, and lighting conditions. Finally, the authors provide guidance for industry, academic, and the public sector with respect to face-recognition technologies.

---

> ### Author Response · Authors · 2022-08-29
> **Following up with Reviewer cekZ**
>
> Thank you again for your thoughtful review. Does our response help address your feedback? We would appreciate the opportunity to engage further if needed.

---

> ### Public Comment · ~Corey_Makowski1 · 2022-12-17
> **Thanks**
>
> Thanks for the review. I also found this website, https://writinguniverse.com/essay-introduction/ and your post while looking for information on how to write a paper. Visit this website if you want assistance with producing a paper. I swiftly and easily wrote my paper with the aid of this website. Some people prefer to go right to the point, but college and university assignments don't operate this way. Students should be clear in their explanations, going logically and slowly from one idea to the next.

---

### Review · Ethics_Reviewer_Cr44 · 2022-08-19

**Recommendation:** 1

**Ethics Documentation:**

The paper provides sufficient detail on their dataset aggregations, the code is available on GitHub and links to the original datasets, and their code is properly licensed on their repo.



**Ethics Review:**

This paper examines the differences in face detection performance between different demographic groups and presents a benchmark to this end, consisting of previously released facial recognition datasets.

The paper includes a proper discussion of licenses and data origination from each existing paper.

Reviewer cekZ's point regarding using "bias against" in this context where facial recognition could potentially cause harm is worth considering. This could help strengthen the ethical discussion of the paper.

---

### Meta-Review · Area_Chair_g4Ho · 2022-09-09

**Recommendation:** Accept
**Confidence:** 4

**Metareview:**

The paper contributes a benchmark dataset to test the robustness to noise of face localization models. It is created by adding a standard set of perturbations to four existing datasets. The paper also includes an analysis of six face detection systems based on this dataset. Since the authors focus on disparities in robustness of the output, and not disparities in performance, it makes for a cleaner set of experiments; on the other hand, it’s a bit more distant from the question that is most relevant from a fairness perspective, which is about performance disparity.

The reviewers appreciated the paper’s contribution to the line of work on auditing facial analysis systems; the scale of the datasets and the experiments; the quality of the writing; and the care that the authors exercised regarding ethical issues. A number of questions and concerns were brought up in the reviews, and the authors have addressed most of them in their responses and revision.

---

### Decision · Program_Chairs · 2022-09-16

Accept